

# Planning spatial sampling of the soil from an uncertain reconnaissance variogram

R. Murray Lark[1], Elliott M. Hamilton[1], Belinda Kaninga[2,3], Kakoma K. Maseka[4], Moola Mutondo[4], Godfrey M. Sakala[2,3], and Michael J. Watts[1]

[1]Centre for Environmental Geochemistry, British Geological Survey, Keyworth, Nottinghamshire, NG12 5GG, U.K.
[2]Zambia Agriculture Research Institute, Mount Makulu, Central Research Station, Lusaka, Zambia
[3]University of Zambia, Great East Road Campus, Lusaka, Zambia
[4]Copperbelt University, Jambo Drive, Riverside, Kitwe, Zambia

*Correspondence to:* R.M. Lark (mlark@bgs.ac.uk)

**Abstract.** An estimated variogram of a soil property can be used to support a rational choice of sampling intensity for geostatistical mapping. However, it is known that estimated variograms are subject to uncertainty. In this paper we address two practical questions. First, how can we make a decision on sampling intensity which is robust, given the uncertainty in the variogram? Second, what are the costs incurred in terms of over-sampling because of uncertainty in the variogram model used to plan sampling? To achieve this we show how samples of the posterior distribution of variogram parameters, from a computational Bayesian analysis, can be used to characterize the effects of variogram parameter uncertainty on sampling decisions. We show how one can select a sample intensity so that a target value of the kriging variance is not exceeded with some specified probability. This will lead to oversampling, relative to the sampling intensity that would be specified if there were no uncertainty in the variogram parameters. One can estimate the magnitude of this oversampling by treating the tolerable grid spacing for the final sample as a random variable, given the target kriging variance and the posterior sample values. We illustrate these concepts with some data on total uranium content in a relatively sparse sample of soil from agricultural land near mine tailings in the Copperbelt Province of Zambia.

## 1 Introduction

When one plans a spatial survey of a soil property by geostatistical mapping, a key choice is the intensity of sampling effort (samples per unit area, or, equivalently, spacing of a regular sampling grid). This decision determines the overall cost of sampling, but also the precision of the predictions and therefore the uncertainty in the resulting information. It has long been recognized that, when the variogram of the target variable is known, either by reconnaissance sampling or from data in a homologous environment, it is possible to compute the prediction error variances (kriging variance) for sampling grids of different intensity and so to find the amount of sample effort that is required to meet a target level of precision of the spatial predictions (McBratney et al., 1981). This approach has been used in practice (e.g., Di et al., 1989; Ruffo et al., 2005). However, it is often the case that the number of observations available from reconnaissance survey are rather limited, which



means that the estimated variogram has considerable uncertainty. This uncertainty in the variogram propagates through the calculation of kriging variance, so the kriging variance achieved by a particular grid spacing has an attendant uncertainty.

Webster and Oliver (1992) suggested that at least 100, and ideally 150–200, data are required to obtain an adequate estimate of the variogram. This raises a difficulty, because for many surveys, 200 data are likely to represent a substantial proportion

of the total affordable sampling effort. It may therefore be difficult to justify the collection of this number of data for sample planning purposes, except in the case of large surveys at national scale. An alternative approach was proposed by Marchant and Lark (2006). This is an adaptive sampling strategy in which data are collected in discrete phases. The distribution of sample points for the initial sampling phase is optimized to minimize uncertainty in variogram parameters, given a wide prior distribution for these. In subsequent phases this initial sample is supplemented to improve the precision of the estimates of

variogram parameters until the uncertainty in the final sample grid spacing required to complete the survey is reduced to an acceptable size. An advantage of this approach is that the total sample effort is controlled by the evaluation of the quality of information available at the end of each phase, and so the information provided by an initial phase with rather fewer than 100 samples can be exploited. We make no arbitrary assumptions about the requisite sample size. A disadvantage, however, is that it is not always feasible to divide a sampling campaign into several phases, particularly if the region to be surveyed is remote,

the samples take a significant time to process and analyse, or the target variable is subject to change over time. Marchant and Lark (2006) suggested that there may, nonetheless, be benefits in simple two-phase optimized designs in which information obtained in the first phase of sampling is used in a numerical optimization procedure to find the distribution of sample points, given those already sampled in phase 1, which minimizes some measure of prediction uncertainty.

These tools for optimization are powerful, but they are computationally costly. Furthermore, they address directly the ques-

tion of how to deploy some specified number of sample points and are cumbersome if the initial question is what survey intensity is required, because this requires a laborious optimization of the distribution of different numbers of sample points. In this paper we consider a simpler approach where the basic procedure of McBratney et al. (1981) is adapted to account for uncertainty in the estimated variogram parameters as quantified in a Bayesian analysis. Such an approach might be useful for making initial decisions on sample effort even if the final distribution of points is optimized using methods such as those of

Marchant and Lark (2007).

In this paper we consider a case study from the Copperbelt Province in Zambia. A reconnaissance survey was undertaken on farmland in close proximity to a mine tailings dam. This survey was based on a nested sampling design (Lark et al., 2017), of which we report on 64 samples which were collected at a site of interest, the land farmed by the inhabitants of Mugala village near Kitwe. This sample size is rather fewer than the minimum of 100 suggested by Webster and Oliver (1992).

One variable measured in this survey was total content of uranium (U) in the soil, and it is that variable which we examine here. Uranium is of interest because the processing of copper ore can produce technically-enhanced naturally-occurring radioactive material (TENORM) in the tailings (residual waste) through the concentration of radionuclides, including uranium. Uranium is known to be a significant constituent of TENORM in wastes produced from copper mines in the Katanga Basin, which includes the Zambian Copperbelt (Katebe et al., 2008).



In this paper we present a Bayesian geostatistical analysis of the data and show how this allows us to quantify the effects of variogram uncertainty on the inferred precision of predictions from sample grids of different spacing, and to support a robust choice of sample intensity.

## 2  Materials and methods

### 2.1  Field work and data collection

The study area was farmland used by the inhabitants of Mugala Village near Kitwe, located in the Copperbelt Province in the north of Zambia, ($12^{\circ}47'16.1''$S $28^{\circ}6'13.2''$E). According to the Exploratory Soil Map of Zambia (1:1 000 000), (Ministry of Agriculture, 1991), the soils at and around Kitwe are mapped as Plateau Soils (legend unit Pu7) which comprises chromi-haplic Acrisols with gleyi-haplic Acrisols and partly skeletal phase dystric Leptosols in the FAO classification (FAO–Unesco, 1974).

A full account of the sampling undertaken in this study is provided by Lark et al. (2017). The sampling was undertaken on transects with sample main stations at intervals of 100 to 200 m. At each main station a soil sample was collected. A second point was collected 100 m from the main station in a direction approximately normal to the direction of the transect. A third sample point was selected 10 m from the second in a random direction and a fourth point 1 m from the third, again in a random direction. The soil samples at each sample point were composites, formed by bulking five cores taken with a Dutch auger (15

15 cm depth of soil from the surface and 5 cm in diameter). These cores were taken from the corners and centre of a square of approximately 30 cm length. The samples were placed in paper sample bags, and a few days after collection each sample was reduced in size by coning and quartering. A total of 64 samples were collected this way at Mugala village.

The soil samples were prepared for analysis in the Inorganic Geochemistry laboratories of the British Geological Survey. Each sample was air-dried and then sieved to pass 2 mm. A sub-sample of the sieved soil was milled to $< 53 \mu$m in an agate

20 ball mill then a 10-g subsample of milled soil was mixed with 3 g of binder and a 32-mm diameter pellet was pressed from this mixture. Total concentrations of a suite of elements were then measured on this pellet by X-Ray Fluorescence Spectrometry (XRFS), wavelength-dispersive XRFS for U. This analysis was done at the XRFS laboratory of PANalytical Ltd., Nottingham U.K.

### 2.2  Data analysis

25 #### 2.2.1  The linear mixed model and its parameters

Summary statistics were computed for the data, and their spatial distribution was examined. The assumption of a stationary mean seemed plausible, so a linear mixed model was proposed for the $n$ data in which they are treated as a realization of a random variable, $Z$. The model takes the form:

$$\boldsymbol{z} = \boldsymbol{\mu} + \boldsymbol{\eta} + \boldsymbol{\varepsilon}, \qquad (1)$$




where $\boldsymbol{\mu}$ is a vector, length $n$, with all values equal to a constant mean and $\boldsymbol{\eta}$ and $\boldsymbol{\varepsilon}$ are mutually independent random effects. The component $\boldsymbol{\eta}$ is distributed as

$$\boldsymbol{\eta} \sim \mathcal{N}\left(\mathbf{0}_n, \xi\sigma^2\mathbf{R}\right) \tag{2}$$

and $\boldsymbol{\varepsilon}$ is distributed as

$\quad \boldsymbol{\varepsilon} \sim \mathcal{N}\left(\mathbf{0}_n, (1-\xi)\sigma^2\mathbf{I}_n\right) \tag{3}$

where $\mathbf{0}_n$ is a vector length $n$ with all elements equal to zero, $\mathbf{I}_n$ is an $n \times n$ identity matrix, $\mathbf{R}$ is an $n \times n$ correlation matrix, $\sigma$ is the overall variance of $Z$ with $\xi\sigma^2$ the variance of the spatially correlated component, $\boldsymbol{\eta}$, and $(1-\xi)\sigma^2$ the variance of the independently and identically distributed element, $\boldsymbol{\varepsilon}$. The spatial correlation is modelled under the assumption of second-order stationarity (Webster and Oliver, 2007) such that element $\mathbf{R}[i,j]$ of the correlation matrix can be modelled as a function of the

lag interval between the $i$th and $j$th observations at locations $\boldsymbol{s}_i$ and $\boldsymbol{s}_j$ respectively. With a small data set we assume that the correlation depends only on the length of the lag vector $h = |\boldsymbol{s}_i - \boldsymbol{s}_j|$. In this study we used the Matérn correlation function (Matérn, 1986):

$$\rho(h) \;=\; \left\{2^{\kappa-1}\Gamma(\kappa)\right\}^{-1} \left(\frac{h}{\phi}\right)^{\kappa} K_{\kappa}\left(\frac{h}{\phi}\right), \tag{4}$$

where $\Gamma(\cdot)$ is the gamma function and $K_{\kappa}$ is a modified Bessel function of the second kind of order $\kappa$. The two parameters

are $\kappa$, a smoothness parameter, and $\phi$, a distance parameter. These have to be estimated in addition to the variance $\sigma^2$ and the parameter $\xi$ which represents the proportion of the variance which is spatially dependent.

Under the linear model with a constant mean the variance parameters $\kappa$, $\phi$, $\sigma^2$ and $\xi$ can be estimated by Maximum Likelihood (Lark, 2000; Zimmerman and Stein, 2010; Diggle and Ribeiro, 2007). It is known that the first of these parameters can be difficult to estimate, and Diggle and Ribeiro (2007) suggest that, rather than estimating it along with the other parameters,

the marginal likelihood is obtained for a set of discrete values of the parameter (i.e. the likelihood maximized with respect to all the other parameters when $\kappa$ is fixed at a specified value). Examination of the profile negative log-likelihood function for the $\kappa$ parameter showed that it was potentially troublesome to estimate. Although the negative log-likelihood was smaller with $\kappa = 2$ than with larger or smaller values, the slope of the marginal likelihood as $\kappa$ was increased above 1.5 was very small. For this reason we followed the guidance of Diggle and Ribeiro (2007) and fixed $\kappa$ at the value for which the profile negative

log-likelihood was smallest, 2.0. All subsequent analyses are conditional on this choice.

Estimates of the other variance parameters were obtained by maximum likelihood. It is possible to quantify uncertainty in these estimates by treating the inverse of the Fisher information matrix as an estimate of the covariance matrix of estimation errors (e.g., Dobson, 1990), but this requires assumptions of linearity which are not plausible for all of the parameters, notably the distance parameter (Marchant and Lark, 2004). An alternative approach is to use a Bayesian formulation of the linear mixed

model under which the variance parameters are treated as random variables with a prior distribution, updated to give a posterior distribution by reference to available data. This has been done in previous studies with the linear mixed model applied to soil data (e.g., Orton et al., 2009; Minasny et al., 2011).



A common way to implement the Bayesian analysis is by Markov chain Monte Carlo (MCMC) methods in which a sample of values from the posterior joint distribution of the variance parameters is drawn. This sample can then be used for further inference about the parameters. Both Orton et al. (2009)l and Minasny et al. (2011) used this approach, and we follow the latter in using the dream algorithm of Vrugt et al. (2009), as implemented in R by Guillaume and Andrews (2012) for the
analysis. For details of this method the reader is referred to Vrugt (2016). In summary the dream algorithm runs multiple MCMC chains in parallel and automatically tunes the 'proposal' distribution, which is used to perturb the values in each chain and explore the parameter space so that the resulting sequence of samples has desirable statistical properties. In dream, and other multichain methods, the perturbation of the chains in each generation is obtained as a combination of a random variate and the difference between the parameters values for randomly selected pairs of chains. In dream computational efficiency
is achieved by updating only random subsets of the parameters in each generation, and by special treatment to identify and manage outlier chains. We used 9 chains in our analysis for 3 parameters, with four chain-pairs used to generate the jump at each sample. Other dream parameters were set at default values which have found to be robust over a range of settings. The first ten percent of values in each chain was discarded to avoid the effects of the burn-in period, which is influenced by initial arbitrary settings. Every tenth output of the chain was selected for the final sample to reduce effects of autocorrelation between
samples. The prior distribution for the variogram parameters was uniform over the admissible range $[0, 1]$ for $\xi$. For the other parameters the priors were uniform for positive values up to a maximum (10 and 1000 respectively for $\sigma^2$ and $\phi$). These latter two priors were judged to be acceptable because the posterior density in each case was very small near the maximum value of the respective parameters.

### 2.2.2   Kriging variance as a random variable

In the analysis undertaken here the smoothness parameter $\kappa$ was fixed at the value selected from the profile likelihood, but the other parameters, $\phi$, $\sigma^2$ and $\xi$, comprise a set which is treated as a random variate:

$$\boldsymbol{\Theta} = \left\{ \Phi, \Sigma^2, \Xi \right\}^{\mathrm{T}}, \tag{5}$$

of which we have $m$ MCMC samples:

$$\boldsymbol{\theta}_i = \left\{ \phi_i, \sigma_i^2, \xi_i \right\}^{\mathrm{T}}, \ \ i = 1, \ldots, m. \tag{}$$

For some $\boldsymbol{\theta}$, and conditional on a specified interval for a square sample grid, $\lambda$, one can compute the kriging variance at the centre of a grid cell. In this paper we consider the ordinary kriging variance. We may write this kriging variance as a function of $\boldsymbol{\theta}$:

$$v_{\mathrm{k}, \lambda} = f\left(\boldsymbol{\theta} | \lambda\right). \tag{6}$$

In our Bayesian formulation of the problem the variogram parameters are treated as random variables, and so we may treat the
kriging variance, conditional on $\lambda$, as a random variable:

$$V_{\mathrm{k}, \lambda} = f\left(\boldsymbol{\Theta} | \lambda\right). \tag{7}$$



We may obtain a sample from this random variable by applying the expression in Eq. (6) to each of the $m$ MCMC samples from $\Theta$:

$$\boldsymbol{v}_{\mathrm{k},\lambda,i} = f(\boldsymbol{\theta}_i|\lambda,), \ \ i = 1,\ldots,m. \tag{8}$$

From this sample we may obtain an estimate of the mean kriging variance, $\bar{v}_{\mathrm{k},\lambda}$ with the specified grid spacing. We can also

compute an empirical estimate of the distribution function for $v_{\mathrm{k},\lambda}$:

$$\widehat{F}(v|\lambda,) = \#\boldsymbol{v}_{\mathrm{k},\lambda}^{\leq v}/m, \tag{9}$$

where $\boldsymbol{v}_{\mathrm{k},\lambda}^{\leq v}$ is the sub-vector comprising all elements of $\boldsymbol{v}_{\mathrm{k},\lambda}$ which are $\leq v$.

We are planning a spatial sampling exercise and wish to ensure that $v_{\mathrm{k},\lambda} \leq v_{\mathrm{t}}$, a target kriging variance. Given the uncertainty in the variogram parameters, the kriging variance achieved by some grid spacing, $\lambda$, is uncertain. One way to deal with this is

to ensure that the probability that the target kriging variance is not exceeded for a chosen spacing is sufficiently large. This is analogous to power analysis for hypothesis testing. If we specify a probability of, for example, 0.8, then we could select a grid spacing, $\check{\lambda}$ such that

$$\widehat{F}\left(v_{\mathrm{t}}|\check{\lambda}\right) = 0.8. \tag{10}$$

We undertook these calculations using the sample of variogram parameters obtained from the MCMC chains as described in

the previous section. The kriging variance for each sample was computed for the cell centre of square sample grids considering spacings up to 300 m.

The target kriging variance, $v_{\mathrm{t}}$, might be selected on various grounds (de Gruijter et al., 2006; Black et al., 2008; Lark and Knights, 2015). For illustrative purposes we chose to select a target kriging variance such that the kriging standard error was 10% of the overall mean concentration. This gives a target kriging variance of $v_{\mathrm{t}} = 0.18$ in this case.

We computed, for each of a set of grid spacings up to 300 m, the estimate of $\widehat{F}(v_{\mathrm{t}}|\lambda)$ and identified the spacing at which this value was 0.8.

### 2.2.3   Tolerable grid spacing as a random variable

The previous section shows how one may select a grid spacing such that the probability that the kriging variance at the centre of a grid cell is less than or equal to a target variance is sufficiently large. We may then wish to quantify the implications of

the uncertainty in terms of the extent to which we are likely to have oversampled to account for it. To do this we introduce the notion of the tolerable grid spacing as a random variable given uncertainty in the variogram parameters.

For some sample from the variogram parameters, $\boldsymbol{\theta}$, it may be possible to find $\lambda_v$, the spacing of a square grid such that the kriging variance at the centre of a grid cell is $v$. This is done numerically, by interpolation between evaluations of the kriging variance for a series of grid spacings and the particular parameter set using Eq. (6). The value $\lambda_v$ is not necessarily defined for

some parameter set, $\boldsymbol{\theta}$. There are two circumstances in which it is not. The first is when

$$v > \sigma^2 + \psi_\lambda, \ \ \forall\lambda, \tag{11}$$



where $\psi_\lambda$ denotes the Lagrange parameter of the ordinary kriging estimate at the centre of a grid cell of spacing $\lambda$. In this circumstance the target kriging variance is so large that it is matched by kriging from the coarsest grid. The second circumstance in which $\lambda_v$ is not defined is when the spatially uncorrelated variance is larger than $v$:

$$v < \xi\sigma^2. \tag{12}$$

In this latter case the kriging variance $v$ cannot be achieved at the centre of the grid cell however fine the spacing. When

$$\xi\sigma^2 < v < \sigma^2 + \psi_\lambda \ \ \forall\lambda,$$

then a unique value of $\lambda_v$ is defined for the $\boldsymbol{\theta}$ and $v$ under consideration, assuming ordinary point kriging or block kriging for a block which is small compared with the grid of observations from which the prediction is made. When $v$ takes a value in the range for which $\lambda_v$ is defined we may express the relationship by a function:

$$\lambda_v = g(v,\boldsymbol{\theta}). \tag{13}$$

We call $\lambda_v$ the tolerable grid spacing, because it is the coarsest spacing compatible with the kriging variance $v$. In the Bayesian approach, where $\boldsymbol{\theta}$ is drawn from a random variable $\boldsymbol{\Theta}$, we can treat the tolerable grid spacing as a random variable $\Lambda_v$

$$\Lambda_v = g(v,\boldsymbol{\Theta}). \tag{14}$$

Consider a situation in which an oracle provides variogram parameters without uncertainty and we can therefore compute

the tolerable grid spacing for target variance $v_{\mathrm{t}}$ without uncertainty. The sample density is $\lambda_{v_{\mathrm{t}}}^{-2}$ samples per unit area. If, in the same circumstances, we did not have access to the oracle, but selected a grid spacing, $\breve{\lambda}$, to achieve Eq. (10), then the oversampling, attributable to our strategy to manage uncertainty would be $\breve{\lambda}^{-2} - \lambda_{v_{\mathrm{t}}}^{-2}$ samples per unit area. This could be directly translated to a cost given analytical costs and logistical costs (e.g., Lark and Knights, 2015). In practice we may use our $m$ MCMC samples of the variogram parameters, $\boldsymbol{\theta}_i$, to compute a set of tolerable grid spacings, where this is defined

$$\boldsymbol{\lambda}_{v_{\mathrm{t}},\lambda,i} = g(v_{\mathrm{t}},\boldsymbol{\theta}_i); \text{ if } \xi_i\sigma_i^2 < v < \sigma_i^2 + \psi_\lambda \ \ \forall\lambda; \ \ i = 1,\ldots,m. \tag{15}$$

One may note the proportion of the MCMC samples for which the tolerable grid spacing is defined. If the numbers of undefined cases are significant then one should note whether this is because the target variance is commonly large relative to the variance $\sigma^2$, as in Eq. (11) or small relative to the uncorrelated variance $\xi\sigma^2$, as in Eq. (12). Either would suggest that a review of the target variance is needed. Otherwise one may compute the corresponding sample densities for each variate in the sample of

tolerable grid spacing and compare these with the sample density used to achieve the condition in Eq. (10). The difference is the oversampling, which, as noted above, can be converted to a cost. If this cost is large then there are two options. The first is to consider whether a larger target variance can be accepted, or a smaller probability that it is achieved. The second is to consider further exploratory sampling to reduce the uncertainty in the variogram parameters. The latter approach may be attractive when a survey is planned over a large region, and additional sampling effort is small relative to the total sample effort

expected.





We followed these procedures, using Eq. (15) to compute a sample of tolerable grid spacings for the target kriging variance of 0.18. We noted the proportion of samples for which this tolerable grid spacing was defined. We computed the corresponding values of the sample density.

We then repeated these calculations assuming that (i) the target kriging variance is 0.18 but that we are prepared to accept a
smaller probability that it is not exceeded (0.75 or 0.70) and (ii) that we can accept a larger target kriging variance of 0.25, but require a probability of 0.8 that it is not exceeded.

## 3   Results

Table 1 shows summary statistics for the data on uranium and their histogram is shown in Figure 1(a). The data are reasonably symmetrically distributed, and there is no evidence from these data that a transformation is required. The empirical variogram
(solid symbols in Figure 2(a)) shows no evidence of a spatial trend. The profile negative log-likelihoods for different values of the $\kappa$ parameter are shown in Figure 2(b). On the basis of these results $\kappa$ was fixed at 2.0. The maximum likelihood estimates of the parameters are presented in Table 2, and the solid line in Figure 2(a) shows the corresponding variogram.

Figure 3 shows the empirical posterior distributions of the three variogram parameters (conditional on fixing $\kappa = 2$) corresponding to 15 282 samples after removing the first 10% for burn-in and then extracting every tenth. Note that the empirical
density is small near the upper prior limit for $\phi$ (1000 m) and for $\sigma^2$ (10). The horizontal bars show the 95% credible intervals for each parameter. Because the distributions are not symmetrical, the credible intervals were the highest density intervals (i.e. for a unimodal distribution the narrowest interval over which the integral of the probability density is 0.95). This was computed using the hdi procedure from the HDInterval package for the R platform (Meredith and Kruschke, 2016).

Figure 4 shows the probability, estimated from the MCMC samples, that the kriging variance at the centre of a square grid
of spacing $\lambda$ m, will be less than or equal to the target kriging variance 0.18. The graph shows the spacing, 44 m (5.12 samples per ha), at which this probability is 0.8. The grid spacing at which the expected kriging variance is equal to the target can be identified from the graph in Figure 5 (solid line), the grid spacing is 75 m (1.78 samples per ha). If one considers only the maximum likelihood estimates of the variogram parameters (Table 2) then the relationship between grid spacing and kriging variance is given by the dashed line in Figure 5, and the target kriging variance is equivalent to a grid spacing of 85 m (1.38
samples per ha).

Treating the grid spacing which achieves the target kriging variance of 0.18 as a random variable, $\Lambda_{v_t}$ (tolerable grid spacing) we found that this was defined for 99.22% of the MCMC samples. Figure 5(a) shows the empirical PDF of the grid spacings, $\lambda_{v_t}$. This distribution is mildly positively skewed with some values in an upper tail, but the mean (86.4 m) and median (84.4 m) are very close. The distribution of corresponding sample densities (rescaled to samples per ha), is strongly skewed, and is
shown on a log scale in Figure 5(b). The median tolerable sample spacing is 1.4 samples per ha, the mean is strongly influenced by the upper tail and its value (5.32 samples per ha) is the 87th empirical percentile of the observations. For this reason we summarize the distribution of tolerable sample densities by its median.





We can now summarize the practical implications of this analysis for further sampling of the soil to map uranium in this environment, or comparable ones. The variogram is uncertain, which is not surprising given the relatively small sample size. Because of this, for any specified sample grid spacing, we cannot be sure about the expected prediction error variance at the centre of a grid cell, because this depends on variogram parameters about which we are uncertain. On the basis of our Bayesian

analysis we can characterize that uncertainty, and so we can compute the probability that the kriging variance for some specified grid cell is below our maximum acceptable value. A strategy to deal with the uncertainty about the variogram parameters is to find a grid spacing such that the probability that our target kriging variance is not exceeded is judged to be sufficient. In this case we find that selecting a sample density of 5.12 samples per ha ensures that the probability the specified kriging variance is not exceeded is 0.8.

The achievement of this level of confidence in the quality of our final spatial predictions comes with a cost; we have to do more sampling than we would if the variogram parameters were known with greater certainty. Our analysis shows that the median-unbiased tolerable grid spacing for the target variance is 1.4 samples per ha (close to what we would have obtained if we used the maximum likelihood estimate of the variogram parameters to find the grid spacing). This means that the median-unbiased oversampling due to the uncertainty is 3.72 samples per ha. This could be a substantial cost for a survey of a large

region.

If this cost is unacceptable then there are three approaches we could take. The first is to allow greater uncertainty that our final predictions are of the target precision. We considered the effect of basing the sampling decision on a smaller probability that the target kriging variance is not exceeded, with the target kriging variance held at 0.18. If we specify a probability of 0.75 rather than 0.8 then the grid spacing increases to 53 m (sample density of 3.54 per ha) and the median-unbiased oversampling

due to uncertainty is reduced to 2.1 per ha. Reducing the probability further to 0.7, the grid spacing is 56 m, and the sample density is 3.13 per ha, with a median-unbiased over sampling of 1.7 per ha.

A second approach is to accept a larger target kriging variance. We considered a value of 0.25. The probability that this is not exceeded is 0.8 with a grid spacing of 110 m (0.82 samples per ha), and the median-unbiased oversampling in this case is 0.38 samples per ha (the tolerable grid spacing was defined for 97% of cases because of the increased probability that the

kriging variance would be bounded at a smaller value than the target).

Both these approaches require that we tolerate greater uncertainty, either in the final predictions (accepting a larger kriging variance) or our level of confidence that the specified kriging variance is achieved. If neither of these is acceptable then we must collect additional data to reduce the uncertainty in the variogram model before planning the final survey.

## 4   Conclusions

In this paper we have introduced two new concepts for use in the planning of a geostatistical survey. In the first we consider the kriging variance achieved with a particular grid spacing as a random variable, given a set of MCMC samples of the variogram parameters. This allows us to select a grid spacing such that the probability that the target kriging variance is not exceeded is met with a specified probability. The second concept is to treat the tolerable grid spacing as a random variable. This allows us



to quantify the effects of variogram uncertainty in terms of expected oversampling. These provide a framework for decision-making about geostatistical sampling based on an uncertain variogram. We have shown that, for some plausible values of the target kriging variance, and required probability that the kriging variance is not exceeded, it may be necessary to accept that considerable over-sampling is necessary. This can be reduced by changing either condition. The scientist planning sampling,

or stakeholders such as regulator or policy makers, can explore the effect of relaxing either the target kriging variance or the probability that it is not exceeded.

An alternative, if the expected oversampling is large, is to put some additional sampling effort into improving the quality of the variogram estimate on which the final survey is to be based. Within the Bayesian framework we cannot compute an expected effect on the parameter uncertainty of including some specified number of additional sample points. However, we

could optimize the distribution of the additional points using the procedures of Marchant and Lark (2006), and if the amount of additional sampling is limited to, for example, 10% of the median-unbiased estimate of oversampling incurred with the uncertainty in the current variogram that will ensure that additional sampling to improve the variogram is not excessive.

In conclusion, in our example we could see that uncertainty in an exploratory variogram can have significant implications in terms of sampling cost under reasonable rules to guide sample planning (in terms of the target kriging variance and the proba-

bility one requires that the target kriging variance is not exceeded under the final sampling design). The concepts introduced in this paper can allow for rational assessment of how far currently available data allow a rational decision to be made on detailed sampling for mapping, and so to make a case for additional sampling before a final survey design is fixed. In summary, the practical guidelines for sample planning are as follows

1. Identify a target kriging variance which represents an acceptable precision for the final spatial predictions of the soil

property of interest.

2. Decide on a level of confidence, expressed in terms of a probability that the target precision is achieved, which is required for sample planning.

3. By a Bayesian analysis of available data obtain samples from the posterior distributions of variogram properties, and from these identify the sample density required to achieve the target precision determined at step (1) with the probability

determined at step (2).

4. Find the distribution of tolerable grid spacings, given the sample of variogram parameters, and from this compute the over-sampling required to achieve the target precision with acceptable probability.

5. If this level of over-sampling is unacceptable then either:

i. Review the decisions made at steps (1) and (2), increasing the acceptable kriging variance or reducing the acceptable

probability of achieving this, or both.

ii. Plan additional sampling to improve the estimate of the variogram. The procedures presented by Marchant and Lark (2006) may be used to plan this additional sampling, but we cannot compute the required additional sampling in



the Bayesian framework. The amount of over-sampling expected with current levels of uncertainty might provide a basis for deciding how much additional sampling to undertake at the reconnaissance stage.

*Competing interests.* The authors have no competing interests to declare

*Acknowledgements.* This paper is published with permission of the Executive Director of the British Geological Survey (NERC). Field

5  and laboratory work for this project was funded by The Centre for Environmental Geochemistry and BGS Global. Author contributions were supported by the UK Department for International Development (DFID) through a Royal Society and DFID Africa Capacity Building Initiative (ACBI) Programme Grant, Award AQ140000. We acknowledge the assistance of staff from ZARI and students at Copperbelt University for help with field sampling. We are grateful to the people of Mugala village for permission to sample on their fields.



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





**Table 1.** Summary statistics on soil uranium content

| | |
|---|---:|
| n | 64 |
| Mean /mg kg$^{-1}$ | 4.29 |
| Median /mg kg$^{-1}$ | 4.25 |
| Standard deviation /mg kg$^{-1}$ | 0.60 |
| Skewness | −0.13 |

**Table 2.** Maximum likelihood estimate of spatial covariance parameters for soil uranium

| | |
|---|---:|
| $\phi$ /m | 55.50 |
| $\kappa^{*}$ | 2 |
| $\sigma^2$ | 0.43 |
| $\xi$ | 0.73 |

$^{*}$ obtained by profile
likelihood



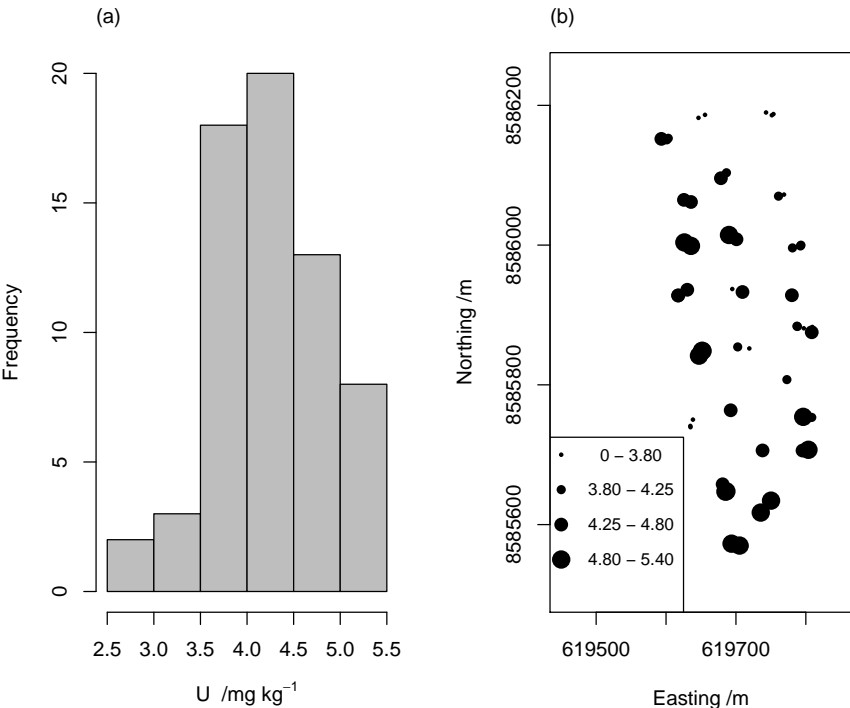

**Figure 1.** (a) Histogram of uranium concentrations in soil samples and (b) post-plot showing the spatial distribution of sample points with symbols indicating concentration intervals delimited by the range and empirical quartiles. The Eastings and Northings are according to the Universal Transverse Mercator projection zone 35.





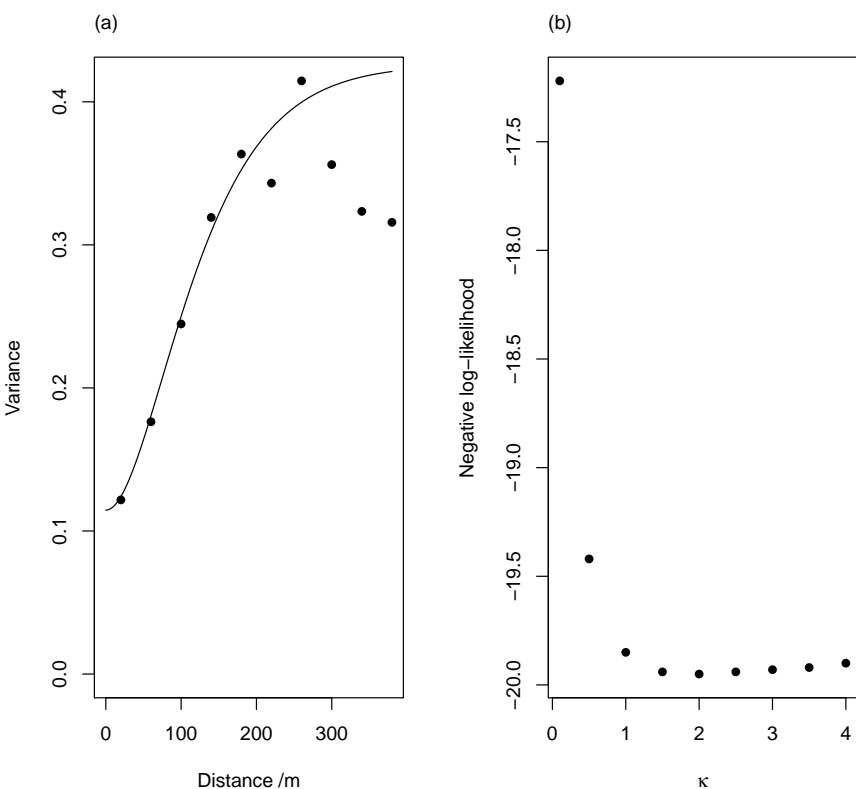

**Figure 2.** (a) Method-of-moments estimates of the variogram of soil uranium content with the maximum likelihood estimate of the variogram superimposed. (b) Profile values of the negative log-likelihood for different values of the $\kappa$ parameter.




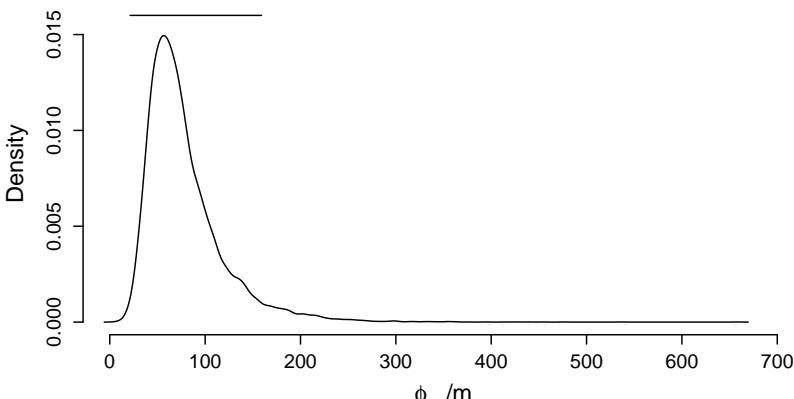

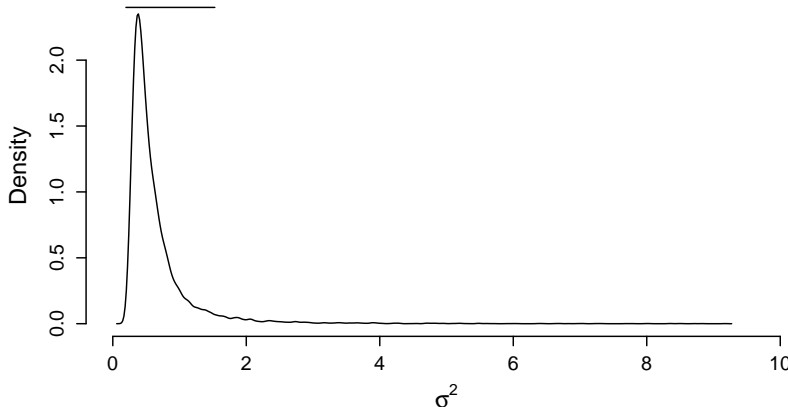

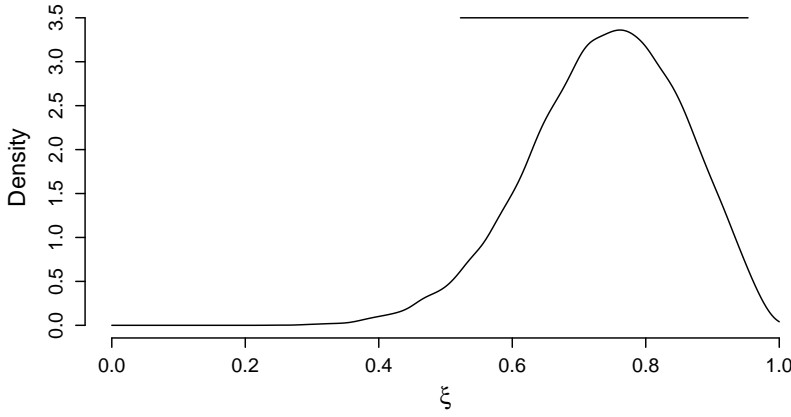

**Figure 3.** Empirical probability density functions for variogram parameters: $\phi$ (top), $\sigma^2$ (middle) and $\xi$ (bottom). These were obtained from the MCMC samples. Horizontal bars show the 95% credible interval for each parameter.

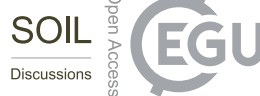

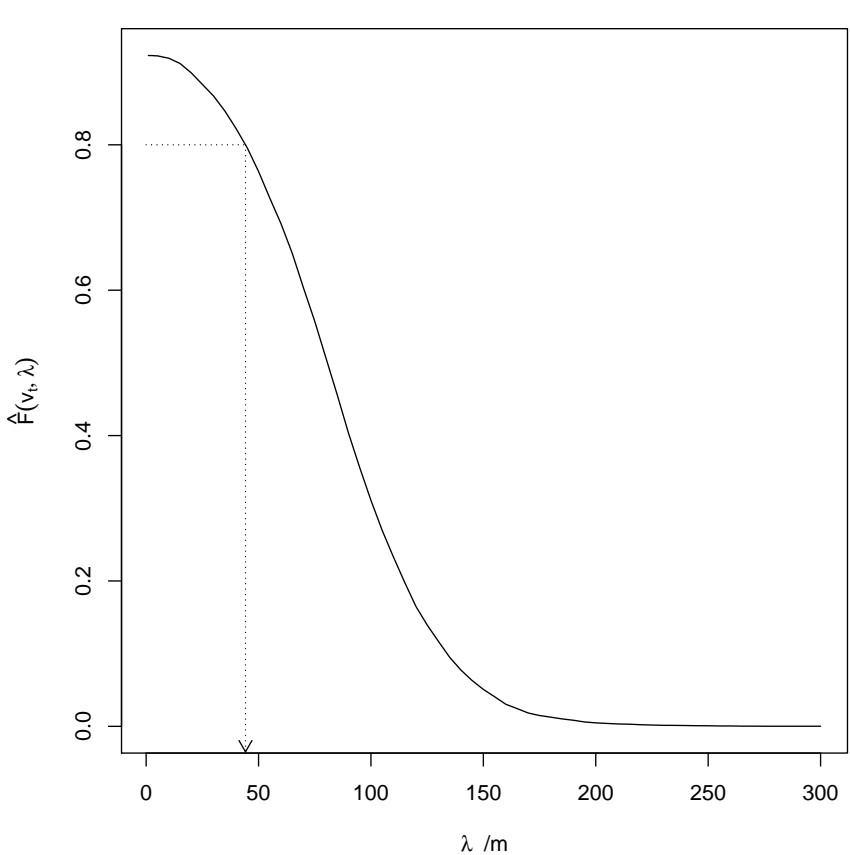

**Figure 4.** Estimates of $\widehat{F}(v\,t|\lambda,)$, Eq. (9) for different grid spacings with $v_{rmt}$ set to 0.18. The dotted line shows the grid spacing (44.2 m) at which $\widehat{F}(v\,t|\lambda,) = 0.8$.



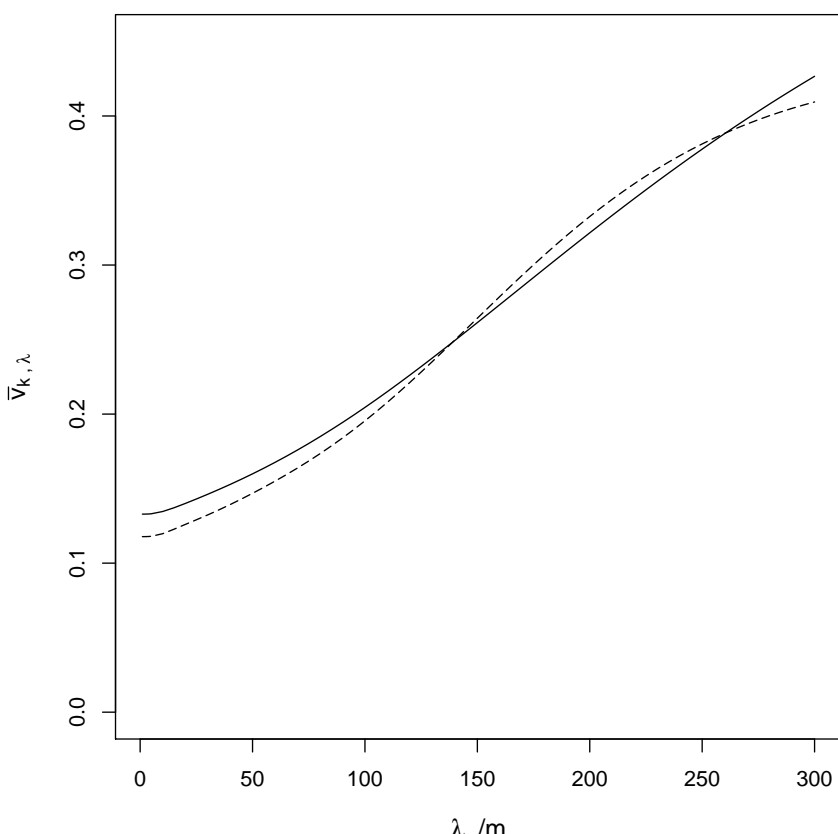

**Figure 5.** Value of kriging variance at the centre of a square grid cell as a function of grid spacing, $\lambda$. The solid line is the expected value of kriging variance over the distribution of parameters, and the dashed line are the values given the maximum likelihood estimates of those parameters.




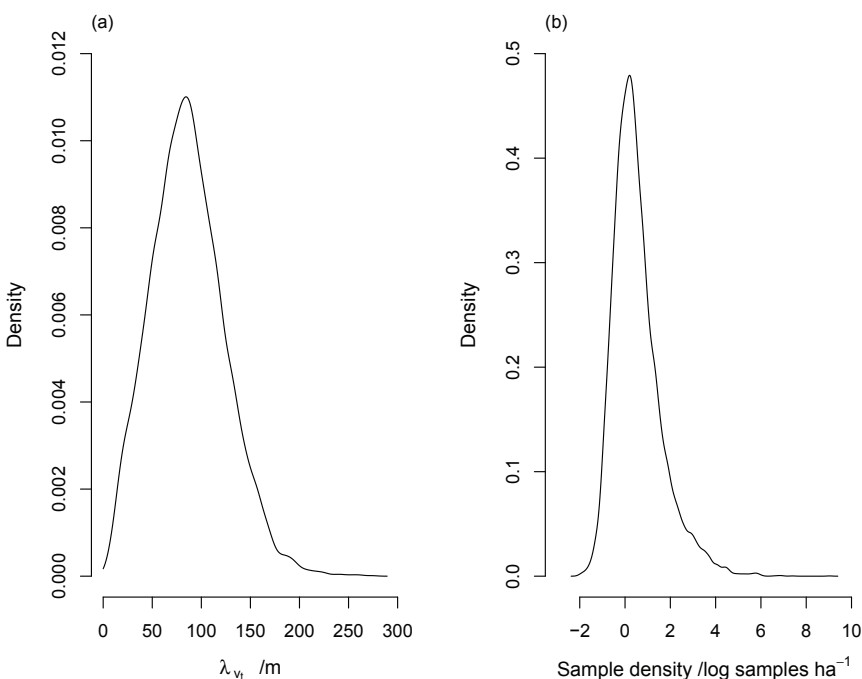

**Figure 6.** Empirical probability density function for grid spacing that achieves the target kriging variance (0.18) and corresponding PDF for the logarithm of the sample density which achieves the same.