# Peer review of "Planning spatial sampling of the soil from an uncertain reconnaissance variogram"

_SOIL, 2017_

## Referee Comment (RC1) · D. Rossiter (Referee) · 11 Sep 2017

David G Rossiter ISRIC-World Soil Information/Cornell University/Nanjing Normal University

General comments

This paper presents a sound approach to planning geostatics-based sampling campaigns based on a small initial set of observations. It shows clearly that conventional methods are expected to seriously underestimate the uncertainty in a proposed second-stage grid sampling. The use of Bayesian methods is appropriate and wellpresented. The paper is a nice mix of rigorous mathematics, well-explained, and the implications for practicioners, as exemplified by the final section of the Conclusions. The expositon is exceptionally clear. Readers need a basis in model-based geostatistics, for which the relevant references are given.

There is a large difference between the required grid spacings as computed by the conventional method (1.4 samples per ha) assumed correct variogram) and the presented method (5.12 samples per ha at 0.8 probability, and even 3.13 at 0.7 probabiity, i.e., still more than double the sampling effort). This is not surprising given the small number of observations, and is well-discussed, but could be highlighted more. The message is that anyone using variogram models estimated by conventional methods from small samples to plan a sampling campaign is seriously underestimating the uncertainty and will likely end up with unacceptable prediction variances. Perhaps this message could be emphasized.

Specific comments

The restriction to regular grid sampling is not explained. Perhaps it could be mentioned that this applies when there is no parts of the target area for which different levels of kriging prediction variance are required.

The presented method assumes first-order stationarity of the target. P3L26 "The assumption of a stationary mean seemed plausible..." is not well-supported. Was this from Table 1, which just shows that the mean and median are similar over the whole dataset? Or from examination of Fig. 1 which does not show any obvious trend? Or from fitting an exploratory trend surface and discovering a poor fit compared to fitting a spatial mean?

Perhaps the authors can briefly mention adaptations that would be needed in the nonstationary case. Would a mixed model first be applied to remove the trend and then the presented method be used on the residual variogram? Or would the parameters of the mixed model also be incorporated into the Bayesian analysis?

P3L9 the use of FAO-Unesco 1974 soil classification is obsolete; I presume this was used because the mentioned exploratory soil map uses it. The named classes have not changed much if at all in the successor World Reference Base (WRB), so this is not a problem for understanding this paper.

P5L17 "These latter two priors were judged to be acceptable..." good, but how were they selected initially? Why the prior range parameter of 1000 m and total variance of 10? Presumably based on expert knowledge of the maximum possible range and total variance under the circumstances, so that larger values need not be considered. But, looking at Fig. 1(b), it's clear that the effective range can not be more than about 200 m (which we see confirmed later in Fig. 3(a)). So why such a large prior upper limit? Similarly, for the total variance parameter $\sigma^2$, Table 1 shows a standard deviation of 0.6 mg kg$^{-1}$; this squared is only 0.36 mg kg$^{-2}$ and indeed Figure 3 (a) shows almost no density past 2 mg kg$^{-2}$. Knowing the statistics in Table 1, why was such a large prior upper limit for the total variance selected? Does this not slow down the MCMC algorithm?

The choice of k=2 from the profile likelihood leads to a higher nugget proportion than the almost-identically likely value 1.5, and even 1 is not much less likely. This in turn leads to higher estimates of kriging prediction variances.

Technical corrections

L3 "a decision on sampling intensity which is robust" the 'which' could refer to either the decision or the intensity; suggest rephrasing "a robust decision on..." Or, if it's the sampling density which is robust, in what sense is that?

Lark et al. (2017) in the text is likely Lark, R.M., Hamilton, E.M., Kaninga, B., Maseka, K.K., Mutondo, M., Sakala, G.M. and Watts M. J.: Nested sampling and spatial analysis for reconnaissance investigations of soil: an example from agricultural land near mine tailings in Zambia, European Journal of Soil Science, 68, In press. This has no year in the reference list.

No year is given for Vrugt J. A.: Markov chain Monte Carlo simulation using the DREAM software package: theory, concepts and MATLAB implementation, Environmental Modelling & Software, 75, 273–316. The text reference implies 2016.

DOI for all entries in the reference list would be useful.

---

## Referee Comment (RC2) · Anonymous Referee #2 · 22 Sep 2017

The authors present an useful extension of an old procedure proposed by McBratney et al (1981) for tuning the sampling effort with regard to a target level of prediction, using a prior knowledge of the variogram of the property to be mapped. The extension lies in enabling the use of uncertain variogram, which, in practices is very often encountered. They propose a nice solution based on a Bayesian approach, which allows propagating the uncertainty on the variogram parameters to the grid spacing to be chosen. The user can make his/her decision considering prior selection of a target kriging variance and of level of risk in exceeding this target variance. It must be noted however that this approach is only valid under the assumption of stationarity mean, which is all the more violated that the size of the study area increases. I do not know if the authors have

enough material for discussing this point but it would at least merit to be recalled in the conclusion, to avoid further misuses of this approach. Beside I invite the authors to extend their approach to another cause of uncertainty of variogram that is as frequent as the lack of sites with exact measurements, and will be more and more encountered in the future. With the emergence of a lot of proxy for estimating some soil properties( soil spectroscopy, resistivimetry, etc...), the problem is less the number of sites than the uncertainty of each property estimation at each site. It seems that the Bayesian approach proposed by the authors could easily be adapted for addressing this case too.

The paper is very clear and well written. It remains some little mistakes that are listed hereafter. Page 2 line 25. this reference does not exist in the reference list. i suppose it is 2006 instead of 2007. Page 3 line 6: It would be useful to provide the size of the study area. Page 8 line 27: Figure 6(a)

---

## Author Comment (AC1) · 16 Oct 2017

We are grateful to Dr Rossiter for his characteristically thorough review. In response to the issues he raises:

- *There is a large difference between the required grid spacings... Perhaps this message could be emphasized.* We agree that more can be said to emphasize the hazards of planning from an uncertain variogram without taking account of the uncertainty. We propose to add brief comments to this effect in the Results and Conclusions sections

[Figure]

- *The restriction to regular grid sampling is not explained.....* We will refer to the possibility of using sample designs which are not regular grids in the Conclusions section of the revised paper.

- *The presented method assumes first-order stationarity of the target....* . The inference that a stationary mean is plausible was based on the post-plot of the data, and on the transitive behaviour of the empirical variogram. We shall make this explicit in section 2.2.1 of the revised paper.

- *Perhaps the authors can briefly mention adaptations that would be needed in the nonstationary case.* . We shall discuss the possible extension of the method to a case with a non-stationary mean in the Conclusions section of the revised paper

- *.. the use of FAO-Unesco 1974 soil classification is obsolete*. The map to which we refer was produced before the introduction of WRB, and so we have to refer to the actual classification on which the map legend was based. We shall make this explicit in section 2.1 of the revised paper.

- *"These latter two priors were judged to be acceptable..."*. Diggle and Ribeiro (2007) make the point that, in the absence of other information on which to base a prior distribution, a robust procedure is to select a uniform prior over sufficiently wide bounds that the posterior density is negligible at the extremes. This is the rationale for what we did, and we can make this specific reference at the end of section 2.2.1. Needless to say one does not select prior parameters on the basis of direct inspection of the data.

- *The choice of k=2 from the profile likelihood leads to a higher nugget proportion...* . We acknowledge the point that the profile likelihood is very flat near the minimum, and that smaller values are not implausible. We agree that the choice of 2.0 rather than a smaller value is likely to lead to slightly larger nugget variances,

and point out that this is therefore conservative (we will have slightly larger kriging variances, other things being equal). We shall add comments to this effect in section 2.2.1.

- Technical corrections

  1. We shall edit the sentence to read 'First, how can we make a robust decision on sampling intensity...'.
  2. Lark et al., 2017. This reference was still in press when the paper was first submitted. All publication details can be put in the revision.
  3. The publication year for this article (2016) will be added.

---

## Author Comment (AC2) · 16 Oct 2017

We are grateful to this reviewer for their careful attention to our paper. In response to the issues they raise:

- We shall make reference to the possibility of dealing with other kinds of variogram uncertainty in the conclusions section of the revised paper.

- Specific corrections

    1. Marchant and Lark (2007) is the correct reference. The publication year was

wrong in the reference list will be corrected.

2. We will give an approximate area for the study region in section 2.1.

3. We shall correct the reference to Figure 6(a).

---

## Author Response (AR1)

**D. Rossiter**.

We are grateful to Dr Rossiter for his characteristically thorough review. In response to the issues he raises:

- *There is a large difference between the required grid spacings... Perhaps this message could be emphasized.* We agree that more can be said to emphasize the hazards of planning from an uncertain variogram without taking account of the uncertainty. See brief comments to this effect in the Results and Conclusions sections at P9, L11–14 and P11, L1–4 in the revised paper.

- *The restriction to regular grid sampling is not explained.....* We refer to the possibility of using sample designs which are not regular grids at P10, L30–32 of the revised paper.

- *The presented method assumes first-order stationarity of the target.... .* The inference that a stationary mean is plausible was based on the post-plot of the data, and on the transitive behaviour of the empirical variogram. We make this explicit in section 2.2.1 of the revised paper (see P3, L27–29).

- *Perhaps the authors can briefly mention adaptations that would be needed in the nonstationary case.* We briefly discuss the possible extension of the method to a case with a non-stationary mean in the Conclusions section (P10, L21–26) of the revised paper.

- *.. the use of FAO-Unesco 1974 soil classification is obsolete.* The map to which we refer was produced before the introduction of WRB, and so we have to refer to the actual classification on which the map legend was based. We make this explicit in section 2.1 (P3, L7–10) of the revised paper.

- *"These latter two priors were judged to be acceptable..."*. At P5, L22–25 of the revised paper we refer to Diggle and Ribeiro (2007) who make the point that, in the absence of other information on which to base a prior distribution, a robust procedure is to select a uniform prior over sufficiently wide bounds that the posterior density is negligible at the extremes. This is the rationale for what we did, and we make this specific reference at the end of section 2.2.1. Needless to say one does not select prior parameters on the basis of direct inspection of the data.

- *The choice of k=2 from the profile likelihood leads to a higher nugget proportion... .* We acknowledge the point that the profile likelihood is very flat near

the minimum, and that smaller values are not implausible. We agree that the choice of 2.0 rather than a smaller value is likely to lead to slightly larger nugget variances, and point out that this is therefore conservative (we will have slightly larger kriging variances, other things being equal). We add comments to this effect in section 2.2.1. (P4, L27–29).

- Technical corrections

  1. We have edited the abstract (P1 L3) to remove the ambiguity .

  2. Lark et al., 2017. This reference was still in press when the paper was first submitted. We have been able to update the reference with full publication details (P12, L18).

  3. The publication year for this article (2016) has been added (P13, L4).

**Reviewer 2**.

We are grateful to this reviewer for their careful attention to our paper. In response to the issues they raise:

- We make reference to the possibility of dealing with other kinds of variogram uncertainty in the conclusions section of the revised paper (P10, L26–29).

- Specific corrections

  1. Marchant and Lark (2007) is the correct reference. The publication year was wrong in the reference list, and is now corrected (P12, L25).

  2. We give an approximate area for the study region in section 2.1. (P3 L6).

  3. We correct the reference to Figure 6(a) at P9 L2 in the revised paper.

[revised manuscript text omitted]